# Improvement of Antialveolar echinococcosis efficacy of novel Albendazole-Bile acids Derivatives with Enhanced Oral Bioavailability

**Chunhui Hu[1,2]\*, Meng Qin[3], Fabin Zhang[1], Ruixue Gao[1], Xuehui Gan[1], Tao Du[1]**

**1** Medical College, Qinghai University, Qinghai, China, **2** StateKey Laboratory of Plateau Ecology and Agriculture, Qinghai University, Qinghai, People's Republic of China, **3** College of Life Science and Technology, Beijing Advanced Innovation Center for Soft Matter Science and Engineering, Beijing University of Chemical Technology, Beijing, China

\* chunhuihu@qhu.edu.cn

**Data Availability Statement:** All relevant data are within the paper and its Supporting Information files.

## Abstract

Alveolar echinococcosis (AE) is a chronic and fatal infectious parasitic disease, which has not been well-researched. Current recommended therapies for AE by the World Health Organization include complete removal of the infected tissue followed by two years of albendazole (ABZ), administered orally, which is the only effective first-line anti-AE drug. Unfortunately, in most cases, complete resection of AE lesions is impossible, requiring ABZ administration for even longer periods. Only one-third of patients experienced complete remission or cure with such treatments, primarily due to ABZ's low solubility and low bioavailability. To improve ABZ bioavailability, albendazole bile acid derivative (ABZ-BA) has been designed and synthesized. Its structure was identified by mass spectrometry and nuclear magnetic resonance. Its physicochemical properties were evaluated by wide-angle X-ray diffraction, differential scanning calorimetry, scanning electron microscopy, and polarizing microscopy; it was compared with ABZ to assess its solubilization mechanism at the molecular level. To avoid the effects of bile acid on the efficacy of albendazole, the inhibitory effect of ABZ-BA on protoscolex (PSCs)s was observed *in vitro*. The inhibitory effect of ABZ-BA on PSCs was evaluated by survival rate, ultrastructural changes, and the expression of key cytokines during PSC apoptosis. The results showed that ABZ-BA with 4-amino-1-butanol as a linker was successfully prepared. Physicochemical characterization demonstrated that the molecular arrangement of ABZ-BA presents a short-range disordered amorphous state, which changes the drug morphology compared with crystalline ABZ. The equilibrium solubility of ABZ-BA was 4-fold higher than ABZ *in vitro*. ABZ-BA relative bioavailability ($F_{rel}$) in Sprague-Dawley (SD) rats was 26-fold higher than ABZ *in vivo*. The inhibitory effect of ABZ-BA on PSCs was identical to that of ABZ, indicating that adding bile acid did not affect the efficacy of anti-echinococcosis. In the pharmacodynamics study, it was found that the ABZ-BA group had 2.7-fold greater than that of Albenda after 1 month of oral administration. The relative bioavailability of ABZ-BA is significantly better than ABZ due to the transformation of the physical state from a crystalline state to an amorphous state. Furthermore, sodium-dependent bile acid transporter (ASBT) expressed in the apical small intestine has a synergistic effect through the effective transport of bile acids.

**Funding:** This research is supported by the science and technology of Qinghai province [No. 2022-QY-201](receiver CH), the Young and Middle-Aged Fund Project of the Medical College of Qinghai University [NO. 2020-QYY-1] (receiver FZ), and the 2022 "Western light" talent training plan of Chinese Academy of Sciences (receiver CH). The funders had no role in study design, data collection and analysis, decision to publish, or preparation of the manuscript.

**Competing interests:** The authors declare no competing competing Interests.

Therefore, we concluded that the NC formulation could potentially be developed to improve anti-AE drug therapy.

## Author summary

Alveolar echinococcosis (AE) is a neglected zoonotic parasitic disease caused by *Echinococcus multilocularis* (*E. multilocularis*). This parasitic infection can cause severe damage to the liver, lungs, and other human or animal organs, which can be fatal if left untreated. The 10-year mortality rate for patients with untreated or improperly treated AE is greater than 90%. Therefore, AE is also called as "worm cancer". Benzimidazoles, including the first-line drug ABZ, are considered the only effective drug against AE. However, ABZ has a low and erratic oral bioavailability due to its low solubility,which has partially led to unsatisfactory clinical efficacy of this oral drug.

In this study, albendazole bile acid derivatives were prepared to improve the anti-*Echinococcus multilocularis* effect. The synergistic effect of steric hindrance and transmembrane transport increased the equilibrium solubility of ABZ-BA *in vitro* by nearly 4-fold and the bioavailability *in vivo* by about 90-fold. The *in vivo* study demonstrated that the pharmacokinetic behavior of the ABZ-BA was superior to that of Albenda. In the pharmacodynamics study, it was also found that the ABZ-BA group had a significant therapeutic effect after 1 month of oral administration. This albendazole bile acid derivative provides an effective and translatable novel approach to oral drug therapy for chronic, fatal, and neglected parasitic infectious alveolar echinococcosis.

## Introduction

Alveolar echinococcosis (AE) is a zoonotic parasitic disease caused by *Echinococcus multilocularis* tapeworm parasites in humans and animals [1]. Almost 100% of the disease occurs in the liver. If untreated, the mortality rate is 90% after diagnosis. AE has the highest mortality rate of all human worm infectious diseases and is often recognized as the "worm cancer" [2]. Over the past 20 years, China (mainly western China) has accounted for 90% of the world's AE disability-adjusted life years (DALYs) [3]. Most of the patients are concentrated in agricultural and pastoral areas and have poor access to medical care. The disease contributes to a cycle of poverty among farmers and shepherds: the poor do not have access to medical care that can treat illnesses, and these illnesses drive them further into poverty. At present, benzimidazole drugs are the only effective drugs against AE, including albendazole (ABZ) [4,5]. However, the solubility of ABZ is low, the crystallization trend is strong, and oral bioavailability is extremely poor ($< 5\%$). Approximately two-thirds of AE patients show moderate improvement after administration, which leads to unsatisfactory clinical therapeutic effects [6].

Compared with other administration routes, oral delivery of anti-*Echinococcus multilocularis* drugs is considered the optimal treatment for patients. At present, the oral preparation of ABZ reported in the literature includes liposome suspension [7], β-Cyclodextrin [8], solid dispersions (SD) [9, 10], and salt formation [11–13]. However, none of the above ABZ preparations have entered the clinical research stage, likely due to their efficacy, development cost, and patient compliance. The liquids are difficult to transport due to their volume and stability risks. β-Cyclodextrin complex and other preparations require many cyclodextrins, which leads to large drug dosages and is difficult to administer. Moreover, drugs with a strong ABZ

crystallization trend are difficult to encapsulate in the hydrophobic cavity of cyclodextrin. Metastable agents such as SD are not an ideal choice for the easily crystallized compound ABZ. Therefore, a novel, effective, stable, and compliant ABZ preparation is needed.

Bile acids (BA) are steroidal triterpenoids synthesized from cholesterol in the mammalian liver and play an important role in lipid metabolism [14]. In the small intestine, BA act as solubilizers to promote the digestion of dietary fat and vitamins, which are then reabsorbed in the terminal ileum epithelial cells through apical sodium-dependent bile acid transporter (ASBT) [15]. In our previous study, we found that ASBT, as a transporter responsible for the reabsorption of most bile acids in the intestine, was expressed in the ileum of rats with hepatic alveolar echinococcosis, and the expression was upregulated compared with the normal group. This means that bile acid can be introduced into ABZ to make albendazole bile acid derivative (ABZ-BA), and therefore increase drug transmembrane transport to increase the oral bioavailability of ABZ [16].

In this study, we hypothesize that the steric hindrance and space-occupying effect of BA could reduce the crystallization of ABZ and improve its solubility [17, 18]. Additionally, specific transport of BA and ASBT could improve the efficiency of ABZ transmembrane transport [19, 20]. If this hypothesis is, the oral bioavailability of ABZ will be greatly improved, significantly improving the anti-hepatic alveolar echinococcosis therapeutic effect of ABZ (**S1 Fig**).

To test this hypothesis, we focused on the solubilization of ABZ-BA and its anti-*Echinococcus multilocularis* effect *in vitro*. First, ABZ and BA were linked by four carbon linkers, 4-amino-1-butanol, using a two-step method. Second, the molecular mechanism of the solubilization effect of ABZ-BA was assessed. Finally, ABZ-BA was demonstrated to have a superior anti-*Echinococcus multilocularis* effect *in vitro* and superior relative oral bioavailability *in vivo* compared with ABZ.

## Materials and methods

### Ethics statement

Rats (male, 200-250g, 6 weeks old) were purchased from Beijing Vital River Laboratory Animal Technology Co., Ltd. (Beijing, China). The animal care and use protocol was reviewed and approved by the Institutional Animal Care and Use Committee (IACUC) at Qinghai University (Approval No. QHDX-2020-0021), in accordance with the Association for Assessment and Accreditation of Laboratory Animal Care (AAALAC) International Animal Care policies.

### Materials

Albendazole (ABZ) and mebendazole (MBZ) were purchased from Ouhe Technology Company, Ltd, (China, batch number 01010119). Bile acid was purchased from MACKLIN Company, (China, batch number C804481). Ltd. 4-amino-1-butanol, N, N-dimethylformamide, Dicyclohexylcarbodiimide, and 4-dimethyl-aminopyridine were purchased from Sigma-Aldrich Company, Ltd. (USA, batch number 178330, 227056, D80002, 39405). Alkaline phosphatase (Elabscience Company, batch number E-bc-k091-m) and Caspase 3 (Elabscience Company, batch number E-ck-a311). Penicillin-Streptomycin solutions were purchased from Procell life science & technology Company, Ltd, (China, batch number WH1021D241). formic acid (HPLC grade), and methanol (HPLC grade) were purchased from Beijing Chemical Works (Beijing, China). Albenda was purchased from GlaxoSmithKline, GSK Company, Ltd, (England, batch number 12240096H).

## Synthesis of Albendazole-Bile acids Derivatives

Albendazole-bile acid derivatives (ABZ-BA) were synthesized in two steps. In the first step, the amino end was linked with the amide of ABZ to form an amide bond; and in the second step, the butanol end was linked with the ester bond of bile acids to form an ester bond. Finally, silica gel was used for separation and purification.

In the first step, 10 g of ABZ were resuspended in 10 mL of 4-amino-1-butanol and 30 mL of N, N-dimethylformamide (DMF) in a round bottom flask, at 90˚C on a warm magnetic stirrer for 24 hours. The reaction process was detected by thin layer chromatography. Subsequently, the reaction solution was slowly added to ice water and stirred at 500 rpm until complete precipitation. After standing for 2 hours, the solution was filtered with a Brinell funnel. The precipitate was collected and dried at 60˚C, resulting in 13.6 g of intermediate crude product.

The crude intermediate product obtained in the first step was mixed with 27.2 g of bile acid (BA), 10.5 g of Dicyclohexylcarbodiimide (DCC), and 1.05 g of 4-dimethylaminopyridine (DMAP) in 30 mL DMF in a round bottom flask and stirred at room temperature for 24 hours. The reaction process was again detected by TLC.

Silica gel was used for separation and purification. The crude separation was performed by dry column loading and dry sample loading, with methanol: dichloromethane = 1:17 as the developing agent. The developing agent was collected and the intermediate, end-product, and by-product components were collected by rotary evaporation. Fine separation was performed to collect the components of ABZ-BA by dry column loading and dry sample loading, with methanol: dichloromethane = 1:30 and 5% acetic acid as the developing agent.

## Nuclear magnetic resonance analysis

ABZ-BA were dissolved in Methanol-D4, and the $^{13}$C NMR and $^{1}$H NMR spectra were obtained at room temperature (Bruker AV-400, Bruker BioSpin GmbH, Rheinstetten, Germany).

## Mass spectrometry analysis

Heated electro-spray ionization (HESI) was paired with a Q-Orbitrap MS in the MS analysis. The flow rate of auxiliary, sheath and sweep gas was set at 35, 10, and 1 (arbitrary unit), respectively. A full MS-ddMS$^2$ mode was used to perform the analysis. The damping gas was in the C-trap and nitrogen was used to stabilize the spraying. Temperatures of 350˚C and 320˚C were set and kept for the auxiliary gas heater and capillary. Under negative mode, 3.0 kV was adopted for the spray voltage, 60 V was used for the S-lens RF level, 50 ms was used for the maximum injection time, and $3.0\ e^6$ was set as the automatic gain control target. Full MS-ddMS$^2$ scan ranged from 150.0000 to 800.0000 m/z. Precise molecular weight [M-H]$^+$ was used for qualitative analysis, the corresponding peak area was used for quantitative analysis, and MS$^2$ fragments were used for further qualitative analysis.

## Determination of equilibrium solubility

An excess amount of ABZ-BA and ABZ were equilibrated in sodium phosphate buffer, pH 7.4, at 37˚C for 48 h. The supernatant was separated from excess solid in solution by ultracentrifugation at 40000 rpm. Subsequently, the supernatant was diluted, and the solution concentration was determined using high-performance liquid chromatography (HPLC) (Shimadzu LC-20AT, Kyoto, Japan). The chromatographic separation was performed with Diamonsil C18 (4.6×150 mm, 5 μm, Waters, Massachusetts, USA). ABZ-BA and ABZ were detected by

ultraviolet (UV) absorbance detection at a wavelength of 240 nm. The mobile phase was methanol /water (70/30 v/v) and the flow rate was 1 mL/min. The injection volume was 20 μL.

## Wide-angle x-ray diffraction (WAXRD)

The ABZ and ABZ-BA powders were characterized by an X'Pert3 Powder X-ray diffractometer (PANalytical, Inc. U.K.), a voltage of 40 kV, and a current of 40 mA. The samples were scanned from $2\theta$ = 5 to 35˚ at a scanning speed of 1˚/min, and the step size was at 0.01˚ $2\theta$ [10].

## Scanning electron microscopy (SEM)

The surface morphology and microstructure of the solid dispersion powder were analyzed using a scanning electron microscope (Hitachi, SU-8010, Japan) operated at an excitation voltage of 20 kV. The samples were mounted on the copper platform and coated with gold for 300 s before observation.

## Polarized optical microscopy (POM)

The morphology of ABZ and ABZ-BA were observed using a Zeiss Axio ImagerA2m microscope.

## Differential scanning calorimetry (DSC)

A Q2000 differential scanning calorimeter (TA Instruments New Castle, USA) was used to measure the melting temperature of all the materials. Approximately 3–7 mg of each sample were weighed and loaded into crimped aluminum pans for DSC ramping at 10˚C/min.

## Protoscolex collection and cultivation

Protoscoleces (PSCs) from *E. multilocularis* were collected aseptically from alveolar echinococcosis obtained from the abdominal cavity of gerbils in Qinghai Research Key Laboratory for Echinococcosis, China. The specific operation steps were as follows: (1) dissect the successfully preserved Echinococcus protoscoleces gerbil model, isolate cysts and wash the residual liver tissue and blood with sterile Penicillin-Streptomycin solution (penicillin (1500 U/ mL) + streptomycin (1000 U/ mL)). (2) Place the clean cysts in sterile glassware, cut them into a slurry, filter the original head section with sterile gauze, and rinse them with Penicillin-Streptomycin solution 3 times. (3) Slightly rotate and shake the filtrate, absorb the suspended connective tissue, impurities, inactivated PSCs, and excess Penicillin-Streptomycin solution in the upper layer, repeat several times and finally concentrate the pure PSCs into the test tube. Viability was assessed by muscular movement (evaluated under an optical microscope), the motility of flame cells, and the exclusion test with methylene blue. The PSCs were incubated in an RPMI 1640 medium containing 0.20 mg/mL liver homogenous serous, 100 U/mL penicillin, 100 μg/ mL streptomycin and 10% calf serum at 37˚C in a 5% $CO_2$ atmosphere at a concentration of (1000–2000) PSCs/mL.

## Detection of protoscolex mortality

The PSCs were divided into blank control groups (Groups 1 and 2), positive control groups (Groups 3 and 4), and experimental groups (Groups 5 to 7). Group 1 was RPMI-1640 medium with 0.20 mg/mL liver homogenate; Group 2 was 0.2% DMSO medium with 0.2% DMSO; Group 3 was RPMI-1640 medium with 0.20 mg/mL liver homogenate and 0.2% DMSO, containing 10 μg/mL Albendazole; Group 4 was RPMI-1640 medium with 0.20 mg/mL liver homogenate and 0.2% DMSO, containing 10 μg/mL albendazole sulfoxide; Groups 5–7 were

RPMI-1640 medium with 0.20 mg/mL liver homogenate and 0.2% DMSO, containing 13.45, 26.9 and 53.8 μg/mL ABZ-BA, with effective ABZ concentrations of 5, 10 and 20 μg/mL, respectively. The culture medium and drug were changed every 3 days for one week. PSCs samples (10 μL) were collected every day and stained with 0.4% trypan blue staining solution, observed under an inverted microscope, and photographed to record PSC mortality.

### Protoscolex microstructure by Scanning Electron Microscopy (SEM)

For ultrastructural studies, 200 μL PSCs were taken from each group 5 days after treatment and fixed with 4% glutaraldehyde in sodium cacodylate buffer at 4°C for 72 hours. Samples were dehydrated by continuous incubation in increasing concentrations of ethanol (50–100%), and ethanol was removed by supercritical extraction. They were then sputter-coated with gold (100 Å thick) and examined on an FEI Quanta 200 scanning electron microscope operating at 20 kV.

### Determination of alkaline phosphatase and Caspase 3 protein activity

0.8mL culture medium was obtained from each group 5 days after treatment, and the content of alkaline phosphatase and Caspase 3 were measured according to the manufacturer's instructions.

### Chromatographic and mass spectrometry conditions

A Dionex Ultimate 3000 RSLC system (Thermofisher) combined with an Accucore aQ C18 column (150 mm×2.1 mm, 2.6 μm, Thermo fisher) was used in the sample separation process. The separation speed was set at 0.3 μL/min for all gradients. 1 μL of injection volume and a constant temperature of (25 ± 1°C were set for the column. The eluents were A, water with 0.1% formic acid (v/v) and B, methanol; the gradient program was as follows: 0–6 min, 8–40% B; 6–8 min, 40–100% B; 8–10 min, 100% B; 10–11 min, 100–8% B; 11–13 min, 8% B.

Heated electro-spray ionization (HE-SI) was paired with a Q-Orbitrap MS in the MS analysis. The flow rate of auxiliary, sheath and sweep gas was set at 35, 10, and 1 (arbitrary unit), respectively. A full MS-ddMS$^2$ mode was used to perform the analysis. The damping gas in the C-trap and nitrogen was used to stabilize the spraying. Temperatures of 350°C and 320°C were set and kept for the auxiliary gas heater and capillary. Under negative mode, 3.0 kV was adopted for the spray voltage, 60 V was used for the S-lens RF level, 50 ms was used for the maximum injection time, and 3.0 $^{e6}$ was set as the automatic gain control target. Full MS-ddMS$^2$ scan ranged from 150.0000 to 800.0000 m/z. Precise molecular weight $[M-H]^+$ was used for qualitative analysis, the corresponding peak area was used for quantitative analysis, and MS$^2$ fragments were used for further qualitative analysis.

### Comparison of systemic pharmacokinetics in SD rats

A cross-over design was employed with male SD rats (n = 7) to evaluate *in vivo* pharmacokinetics. The rats were subjected to fasting for one night before each administration and were fed 4 hours after administration. ABZ was administered by gavage (25 mg/kg). Blood was collected under isoflurane anesthesia at a flow rate of 2.0 mL/min. At 0.25, 0.5, 1, 2, 2.5, 3, 4, 6, 8, 12, and 24 hours after administration, 200–500 μL blood was collected from the orbital venous plexus and centrifuged at 13000 rpm for 5 min. The upper plasma was stored at—80°C for testing.

Plasma samples were thawed at room temperature, and 100 μL was added to an equal volume of mass spectrometry grade methanol, vortexed for 1 min, and centrifuged again at 13000

rpm for 5 min. 100 μL of the supernatant were added to 100 μL of 200 ng/mL mebendazole IS solution, vortexed for 1 min, and centrifuged again at 13000 rpm for 5 min. The resulting sample was injected by UHPLC-HR-ESI-MS with an injection volume of 1 μL. The pharmacokinetic study of SD rats was performed according to the standards recommended by the "Guidelines for the Care and Use of Laboratory Animals" (Animal Laboratory Resources Institute, 1995) and was approved by the Institutional Animal Care and Use Committee of Qinghai University.

## Establishment of in situ models of secondary hepatic alveolar echinococcosis in rats

Six-week-old mice weighing 200-250g were purchased from a market (Weitong Lihua Laboratory Animal Technology Co., Ltd, Beijing, China). The rats were injected with protoscolex (PSC) suspension (1000–2000) PSCs/mL×0.1 mL/each using the open-abdominal direct liver puncture method. The modeling quality was confirmed by Mindray's Resona7 color Doppler ultrasound system. The lesion volume was calculated by the length (X) and width (Y) of its ultrasound transverse section and the width (Z) of its longitudinal section, X·Y·Z. Rats with relatively uniform lesion volumes were selected as model animals for further experiments (Fig 1).

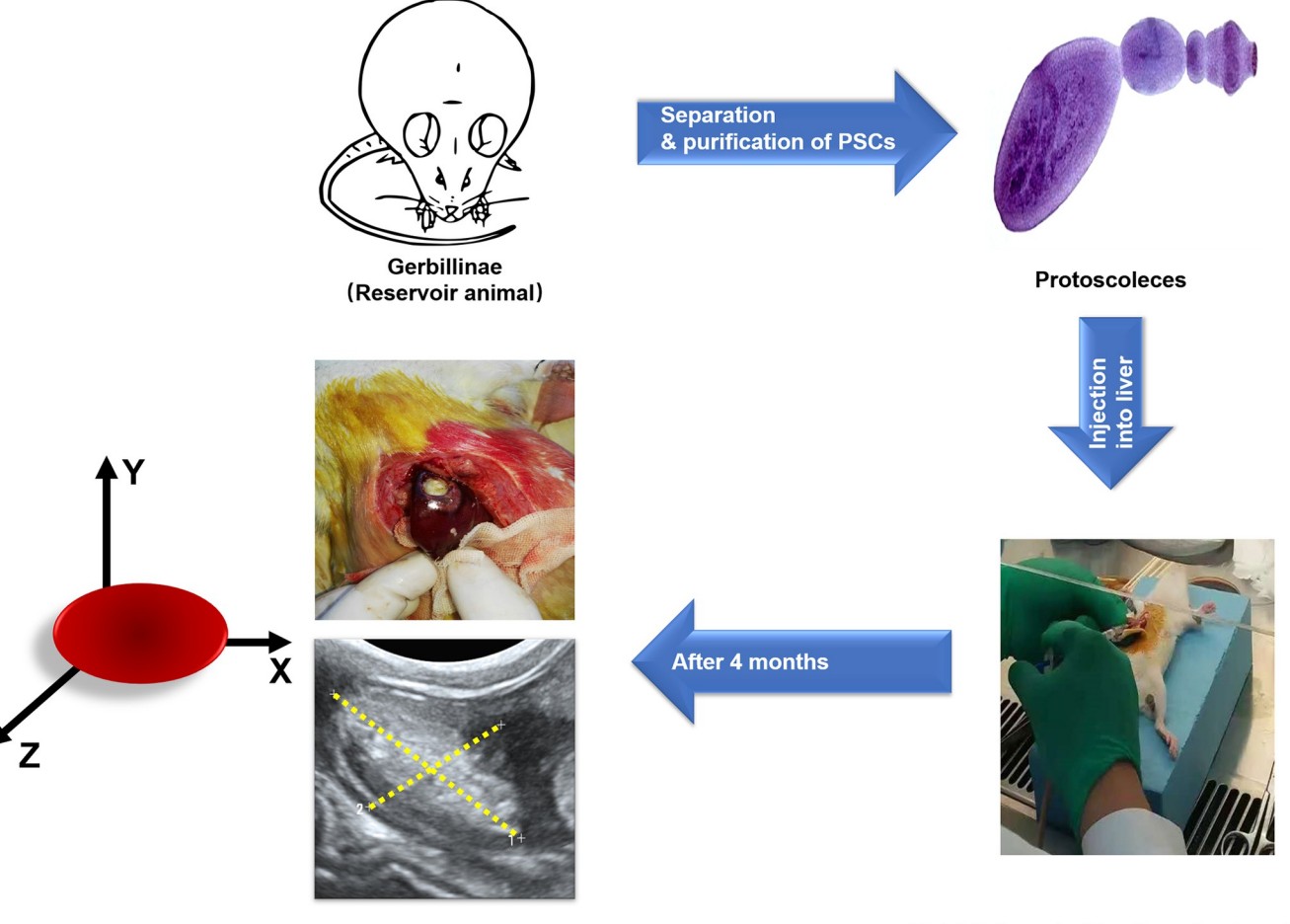

**Fig 1. The establishment of hepatic alveolar echinococcosis and lesion structure in model rats.**

All animal-related experiments were approved by the Animal Care and Utilization Committee (IACUC) of Qinghai University. The animals were housed in a room with controlled temperature ($25 \pm 2°C$) with light-cycled (12 h light/dark cycle). Food and water were provided ad libitum.

### *In vivo* therapeutic efficacy in an in-situ AE animal model

At the 16th week after establishing an experimental secondary AE in rats, we divided the model rats into three groups. The first is the control group, with six infected rats were received with CMC-Na of 0.9% saline. In the second group, rats were received with 0.2 mL of Albenda suspension. In the third group, rats were received with 0.2 mL of ABZ-BA suspension. The three groups were administered by gavage with the dosage of 25 mg/kg/d, which was calculated according to the equivalent dose convert ratio between humans and animals based on the surface body area of humans. After a 30-day, once-daily oral administration, the liver tissue lesions were obtained and weighed to evaluate the cyst inhibition rate after one month of administration.

### Histological observation

Some peripheral regions of infected liver tissue were fixed in 10% neutral buffered formalin at 4°C for 48h, followed by dehydration, paraffin embedding, cut into 2–5μm thick sections, and hematoxylin-Eosin stained for observation.

### Scanning electron microscope (SEM)

Samples were fixed with 4% glutaraldehyde for 72h at 4°C, dehydrated in graded 50–100% ethanol, and gold-plated (100 Å thick) after removal of the ethanol by supercritical extraction. The microstructure of the echinococcosis lesion was examined on an FEI Quanta 200 scanning electron microscope at 20kV.

### Statistical method

SPSS 22.0 was used for statistical analysis. The measurement data of normal distribution was expressed by x $\pm$ *s*. One-way ANOVA was used for inter-group comparisons, and the LSD t-test was used for multiple comparisons. $P < 0.05$ means the difference is statistically significant.

## Results and discussion

### Design and Synthesis of Albendazole-Bile acids Derivatives

Based on previous experimental results, we prepared ABZ-BA using a two-step reaction. Using 4-amino-1-butanol as the linker in the first step, the amino end was linked with the amide of ABZ to form an amide bond; in the second step, the butanol end of the linker was linked with the ester bond of bile acids to form an ester bond. The final product, ABZ-BA, was obtained through crude and fine separation of silica gel column chromatography, as shown in **Fig 2**.

The theoretical m/z of ABZ-BA is 713.4306, and the measured m/z of the sample was 713.4311. These test results are within a reasonable range, suggesting the sample was pure (**Fig 3A and 3B**). As shown in the $^1$H-NMR spectrum, $H_{1-4}$ is the characteristic hydrogen of ABZ, and $H_{5-11}$ is the characteristic hydrogen of BA. The above can be found in the $^1$H spectrum of the sample. At the same time, the ratio of 11:2:3 is about 1:1:1, indicating that ABZ and BA are linked in a 1:1 ratio (**Fig 3C**). The feature C of linkers $C_{1, 16, 17}$ was identified by $^{13}$C-NMR

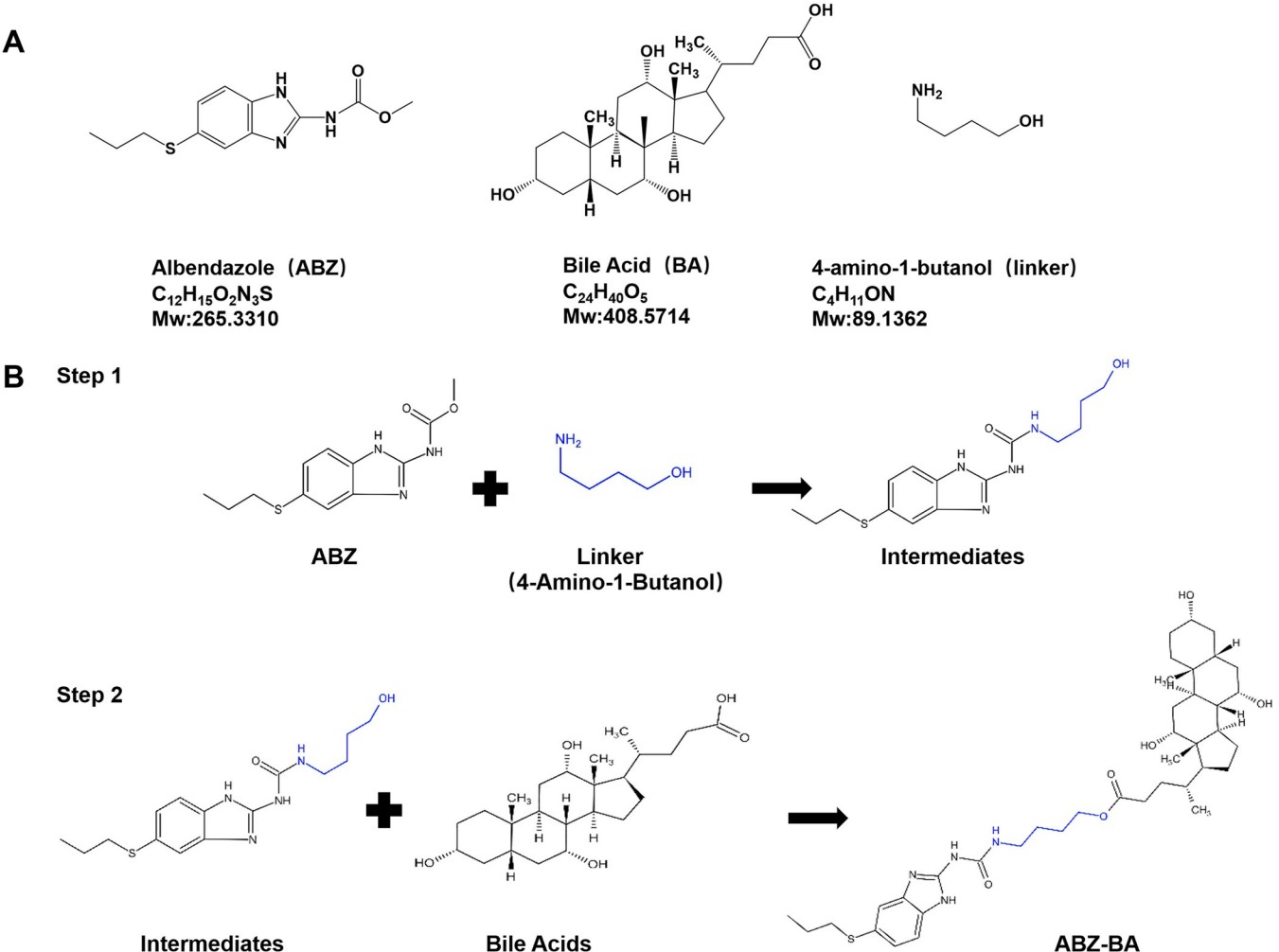

**Fig 2. Structure, physicochemical properties, and synthesis route of ABZ-BA.** (A) Structure of drug and ABZ-BA albendazole, bile acid and 4-amino-1-butanol. (B)ABZ-BA was synthesized in two steps with 4-amino-1-butanol as a linker.

spectrum analysis; and the ABZ feature C-linker $C_{2, 7, 8, 14}$ along with BA C-linker $C_{9-17}$ could also be found in the spectrum. Therefore, ABZ-BA linkage was demonstrated, with the expected ABZ and BA fragments and C1,16 linker ester and amide bonds (**Fig 3D**).

According to WHO treatment guidelines, patients with alveolar echinococcosis must take ABZ orally for extended periods, but the strong crystallization trend of ABZ leads to low solubility and poor oral bioavailability, which seriously limits its clinical application. We previously confirmed that the expression of ASBT in the ileum of a rat model with alveolar echinococcosis is significantly higher than that of normal rats. Therefore, a new formulation ABZ (ABZ-BA) chemically linked to bile acids, the ligand of ASBT, should greatly improve drug oral absorption.

## Solubilization mechanism of albendazole bile acid derivatives

The same amount of ABZ-BA and ABZ was dissolved in PBS solution and dissolved using the same treatment method. High-performance liquid chromatography (HPLC) test results showed that the solubility of ABZ-BA was about 12 times higher than that of ABZ. By

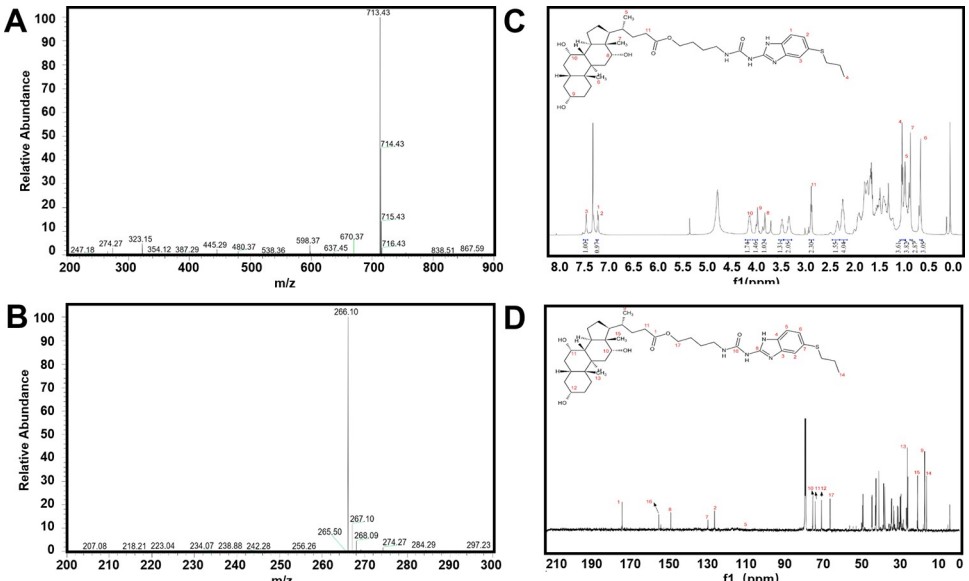

**Fig 3. Structural identification of ABZ-BA.** (A) High-resolution mass spectrum of ABZ-BA. (B) High-resolution mass spectrum of ABZ. (C) ABZ-BA $^1$H-NMR spectrum. (D) ABZ-BA $^{13}$C-NMR spectrum.

comparing the content of ABZ, the solubility of ABZ in ABZ-BA (expressed by Abz later) was about 4 times higher than ABZ, which was statistically significant (**Fig 4A**).

According to the wide-angle X-ray diffraction (WAXRD), ABZ exhibited characteristic crystalline peaks at $2\theta$ values of 7.18˚, 11.43˚, 14.16˚, 18.04˚, 24.86˚, and 30.68˚. In contrast, the diffraction pattern of ABZ-BA was irregular, without characteristic diffraction peaks (**Fig 4B**). These results show that ABZ is crystalline, while ABZ-BA is amorphous. This confirms that Bile acid successfully inhibits ABZ crystallization, and ABZ changing from a crystalline state to an amorphous state could be the primary reason for this increase in solubility.

Crystalline drugs have a clear melting point, while amorphous drugs have a glass transition temperature ($T_g$) without a melting point peak. Based on the above theory, we successfully measured the melting point of ABZ by differential scanning calorimetry (DSC), resulting in an endothermic peak of 216.11˚C (**Fig 4C**). ABZ-BA has no endothermic peak above 200˚C, but there is an obvious step endothermic phenomenon at about 79.12˚C, which is a typical endo-thermic peak of $T_g$. This further proves that ABZ-BA exists in an amorphous state.

Scanning electron microscopy (SEM) results show that ABZ is a flat sheet crystal and ABZ-BA is an irregular sheet (**Fig 4D**), further proving the change of ABZ-BA from a crystal structure of ABZ. Similarly, polarizing microscopy (POM) showed ABZ as a sheet crystal with strong refraction. In contrast, no refractive crystal was observed with ABZ-BA (**Fig 4E**). This demonstrates that ABZ-BA changes the crystallization trend of ABZ by increasing steric hin-drance, and changes from a crystalline to an amorphous drug.

### *In vitro* study of ABZ-BA against *Echinococcus multilocularis*

Changes in the survival rate of the protoscolex (PSCs) after 7 days of treatment in different experimental groups are shown in **Fig 5A**. There was almost no change in the survival rate of PSCs in the liver homogenate group (Group 1) and the vehicle DMSO group (Group 2). ABZ positive control group (Group 3), albendazole sulfoxide (ABZ-SO, Group 4) and ABZ-BA groups (Groups 5 to 7) showed time and concentration-dependent inhibitory effects on PSC.

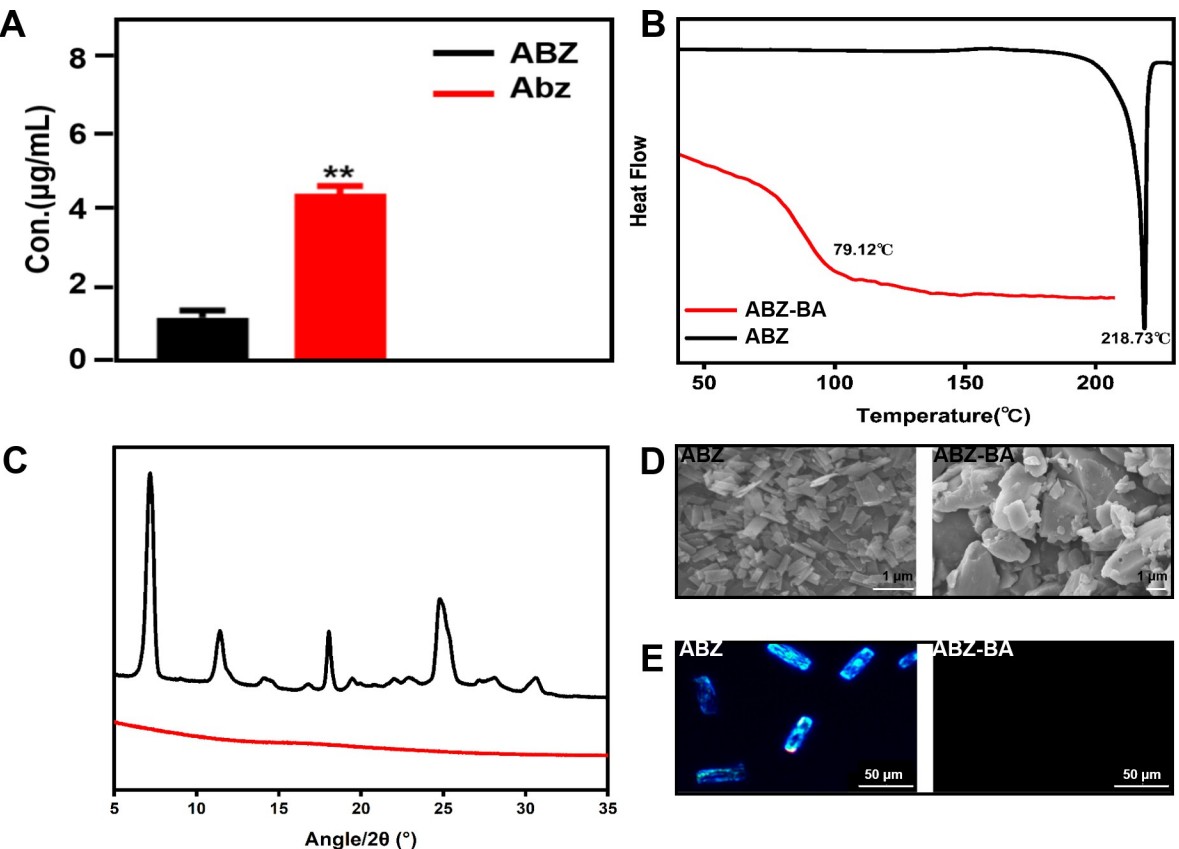

**Fig 4. Study on the equilibrium solubility and solubilization mechanism of ABZ-BA.** (A) Equilibrium solubility of Abz and ABZ (n = 3), where Abz represents ABZ in ABZ-BA, **P<0.001, compared with the ABZ group. (B—E) The thermal analysis characteristic spectra, wide-angle X-ray diffraction (WAXRD) spectra, scanning electron microscope (SEM) spectra, and polarizing microscope (POM) spectra of ABZ-BA and ABZ, respectively. SPSS 22.0 was used for statistical analysis. The measurement data of normal distribution was expressed by x ± s. Student's t test was used for inter-group comparisons. P < 0.05 means the difference is statistically significant(**p<0.01).

Refer to **Table 1** for detailed grouping and composition of culture medium. It should be noted that the effective component of ABZ against echinococcosis is ABZ-SO, which is mainly bio-catalysts by CYP3A4 enzyme in the liver *in vivo*. Therefore, we added the liver homogenate group as a blank control in the experimental design to evaluate the effect of liver homogenate on survival rate. At the same time, the positive control group was set as the ABZ group and ABZ-SO group to evaluate the effect of the active component ABZ-SO against PSCs, and to simulate the effect of ABZ and ABZ-BA on PSCs *in vitro*.

We next observed the microstructure of PSCs using an optical microscope (OM) and a scanning electron microscope (SEM) 5 days after treatment (**Fig 5B and 5C**). The morphology of two blank control groups (DMSO and liver homogenate) was close to normal, and the top three suckers were visible. In ABZ and ABZ-SO groups, the mortality of PSCs was significantly higher than in the blank groups. Under the OM, the PSCs became vacuolar and transparent. Under the SEM, the PSCs elongated and curled up. ABZ-BA groups showed obvious shrinkage at the lowest concentration, but the suckers were still clearly visible. PSCs from intermediate and high concentration groups were strikingly distinct from their original form, appearing in a disc state after death.

Alkaline phosphatase (ALP) is considered an index of PSC damage. Therefore, ALP changes in the supernatant of the PSC culture medium were measured 5 days after treatment

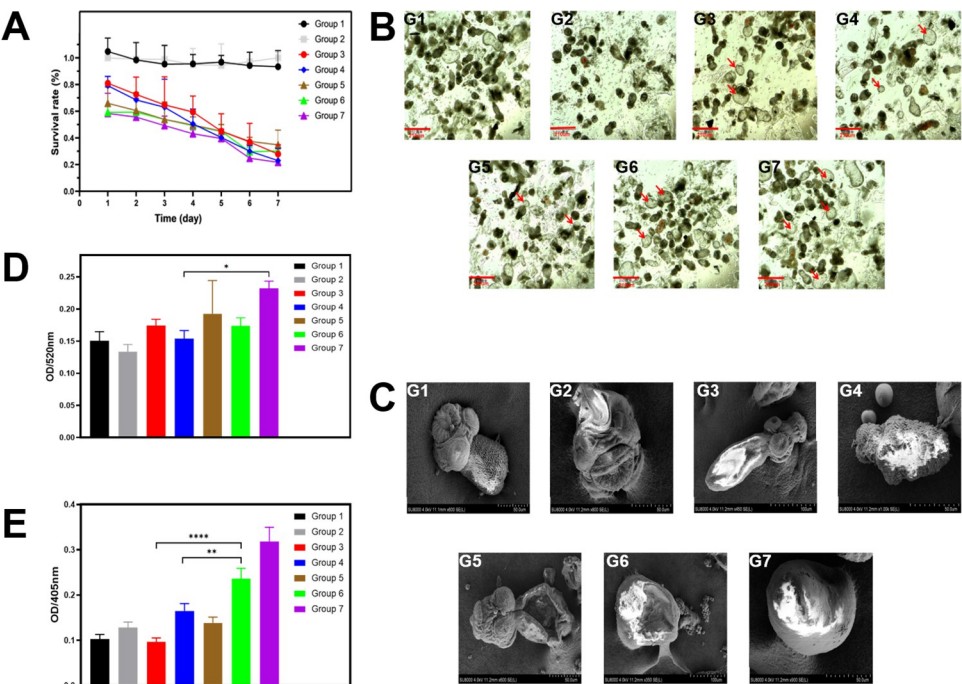

**Fig 5. *In vitro* study on the anti-PSC effect of ABZ-BA.** (A) *In vitro* survival rate of PSCs (n = 3). (B) Representative OM images of PSCs after treatment with different formulations for 5 days (red marked as dead PSC). (C) Representative OM images of PSCs after treatment with different formulations for 5 days. (D) ALP expression level in PSCs culture supernatant. (E) The expression level of caspase 3 in PSCs culture supernatant. (*: P<0.05). SPSS 22.0 was used for statistical analysis. Normally distributed measurement data was expressed by x± s. One-way ANOVA test was used for inter-group comparisons, and Tukey's t-test was used for multiple comparisons. P < 0.05 means the difference is statistically significant(*p<0.05, **p<0.01).

(Fig 5D). There was no significant difference between the same concentration of ABZ-BA and the positive control group, indicating that the efficacy of ABZ-BA was identical to that of ABZ and ABZ-SO. Caspase-3 is an important apoptosis index and was therefore measured in the supernatant of PSCs culture medium 5 days after treatment (Fig 5E). Compared with the positive control group, the content of caspase 3 in the ABZ-BA group was significantly higher, suggesting a superior effect of ABZ-BA on PSCs apoptosis. The ABZ-BA apoptotic effect was concentration-dependent.

**Table 1. Grouping of ABZ-BA and PSC *in vitro* co-culture and composition of culture medium.**

|  | Solution composition |
| --- | --- |
| Group 1 | liver homogenate |
| Group 2 | 0.2%DMSO+ liver homogenate |
| Group 3 | 10μg/mL ABZ+0.2%DMSO+ liver homogenate |
| Group 4 | 10μg/mL ABZ-SO+0.2%DMSO+ liver homogenate |
| Group 5* | 13.45μg/mL ABZ-BA+0.2%DMSO+ liver homogenate |
| Group 6* | 26.9 μg/mL ABZ-BA+0.2%DMSO+ liver homogenate |
| Group 7* | 53.8 μg/mL ABZ-BA+0.2%DMSO+ liver homogenate |

*:The high, medium and low concentration ABZ-BA group contains 5 μg/mL, 10 μg/mL and 20 μg/mL ABZ respectively.

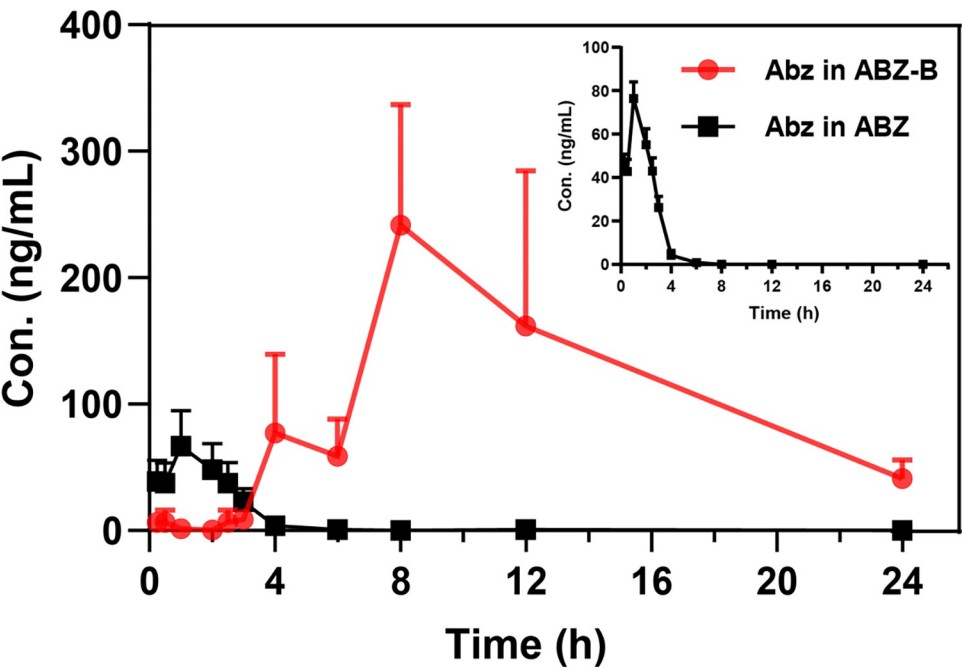

**Fig 6. Drug release behavior of ABZ-BA and ABZ in vivo (n = 7).**

## Comparison of systemic pharmacokinetics in SD rats

The drug pharmacokinetics of ABZ-BA and ABZ were assessed in SD rats at 25 mg/kg oral administration. Since 2.69 g ABZ-BA is equivalent to 1 g ABZ, the ABZ concentration in the plasma of the ABZ-BA group must be calculated from the ABZ-BA concentration detected by UHPLC-MS. Abz is used to represent the ABZ content in ABZ-BA below. The drug-time curve and pharmacokinetic parameters are shown in **Fig 6** and **Table 2**. The data of standard curve (**S1 Table**), precision, accuracy (**S2 Table**), Extraction recovery, Matrix effect (**S3 Table**) stability (**S4 Table**) and specificity (**S2 Fig**) show that the established UHPLC-MS method is feasible.

   We next performed a pharmacokinetic study in SD rats, with an oral administration dose of 25mg/kg of active product ingredient (API). The blood concentration of ABZ was lower

**Table 2. Pharmacokinetic parameters of ABZ-BA and ABZ in SD rats (n = 7).**

| Parameter | | Abz* | ABZ |
|---|---|---|---|
| $AUC_{(0-24)}$ | (mmol/L/h) | 2391.42 ± 792.34 | 91.50 ± 42.66 |
| $AUC_{(0-\infty)}$ | (mmol /L/h) | 3836.20 ± 2796.73 | 101.02 ± 45.91 |
| $MRT_{(0-24)}$ | (h) | 4.12 ± 0.12 | 11.62 ± 0.91 |
| $MRT_{(0-\infty)}$ | (h) | 10.40 ± 12.83 | 13.94 ± 3.98 |
| $t_{1/2z}$ | (h) | 17.27 ± 6.94 | 5.51 ± 4.27 |
| $T_{max}$ | (h) | 9.14±1.95 | 10.86 ± 1.95 |
| $V_{z/F}$ | (L/kg) | 0.02 ± 0.01 | 1.99 ± 1.62 |
| $C_{max}$ | (mmol/L) | 279.03 ± 57.95 | 8.44 ± 3.60 |
| $F_{rel}$* | | 2613% | |

*: Abz is the pharmacokinetic parameters of ABZ-BA active drugs

*: Frel (relative bioavailability) = $AUC_{Abz(0-24)}/AUC_{ABZ\ (0-24)}$

than the lower limit of detection by 1ng/ml after oral administration for 8 hours, while ABZ-BA can be easily detected (**Fig 6**). Comparing the amounts of Abz and ABZ, the $C_{max}$ of ABZ was (76.39 ± 7.62) ng/ml, while the $C_{max}$ of Abz was (280.34 ± 58.23) ng/ml, significantly higher than that of ABZ. Moreover, the $AUC_{0-24}$ of Abz is 26.13 times that of ABZ, suggesting that the relative bioavailability ($AUC_{Abz}/AUC_{ABZ}$) of the drug increased nearly 26-fold.

At the same time, there was a "double-peak" phenomenon in the drug time curve of Abz, with 17.27 hours half-life, and Abz was still detected 24 hours after administration. This could be due to the influence of "enterohepatic circulation" [21,22]. When oral drugs are absorbed into the blood through the small intestine, rapid blood circulation mixes the drugs into the whole blood within a few minutes, and the drugs immediately penetrate the tissue from the capillaries. At the same time, the drug is removed from the blood mainly through liver and kidney clearance and is eliminated in a constant form and/or as a metabolite of the original drug. Some drugs cleared by bile can be reabsorbed from the gastrointestinal tract into the systemic circulation and excreted into the bile again, resulting in multiple peaks in the drug time curve and significantly prolonging clearance half-life. This process is called "enterohepatic circulation" [19,23,24]. The specific recognition of bile acid by ASBT can effectively increase the reabsorption of ABZ-BA, promote enterohepatic circulation, and further increase drug bioavailability.

## Pharmacodynamic evaluation of anti-hepatic alveolar echinococcosis

To examine how ABZ preparations affect secondary hepatic alveolar echinococcosis *in vivo*, we performed ultrasonography on model rats. Our results showed an extremely irregular inner wall structure and uneven anechoic, resulting in poor internal sound transmission in liver lesions. The boundary between the surrounding parenchyma and the liver parenchyma was unclear, showing enhanced calcification deposits and posterior echoes. The solid part of the lesions in other parts had small punctate calcifications, with obviously attenuated echoes in the back. After 4 weeks of administration, the lesion volume in the ABZ-BA and Albenda treatment groups was 0.35±0.18 cm³ and 0.95±0.25 cm³, respectively ($p<0.01$). The anti-alveolar echinococcosis results showed the same trend as *in vivo* pharmacokinetics (**Fig 7A–7D**) because most antiparasitic drugs are concentration-dependent: the higher the concentration, the better the pharmacodynamics [25,26].

*Echinococcus* tapeworms parasitize the liver and cause lesions to form cysts. The cyst wall is composed of two layers, of which the inner layer (also known as the germinal layer) is directly surrounded by cystic fluid, and the outer layer is the layer secreted by the inner layer, which is composed of a relatively tough laminar structure without cells [27]. It has been reported that ABZ can lead to a loss of cytoplasmic microtubules of vesicles and enterocytes, and the redistribution of cytoplasmic vesicles and other organelles in worms (especially Echinococcus species) [27]. The insecticide ABZ inhibits the assembly of microtubules by binding to the colchicine-sensitive site of β-tubulin in the intestinal cells of worms. The digestive enzymes released by tubulin can damage tissue cells (especially the germinal layer). Therefore, the efficacy of ABZ can be determined by the microstructure of the cyst.

Hematoxylin-eosin stained histological images demonstrate differences in cyst walls between the untreated and treated groups. In the untreated group, the laminar edge of the cyst wall was visible, the thickness was uniform, and the germinal layer was dense and flat, which was closely connected with the lamination. However, after Albenda treatment, the space between the coating layer and the germinal layer was slightly enlarged and part of the germinative layer fell off, although the structure of the coating layer was relatively complete. The TABZ-HCl-H treatment resulted in the dissolution of some laminate layers or severe edge damage, and the complete peeling of the hair-germ layer (**Fig 7E**).

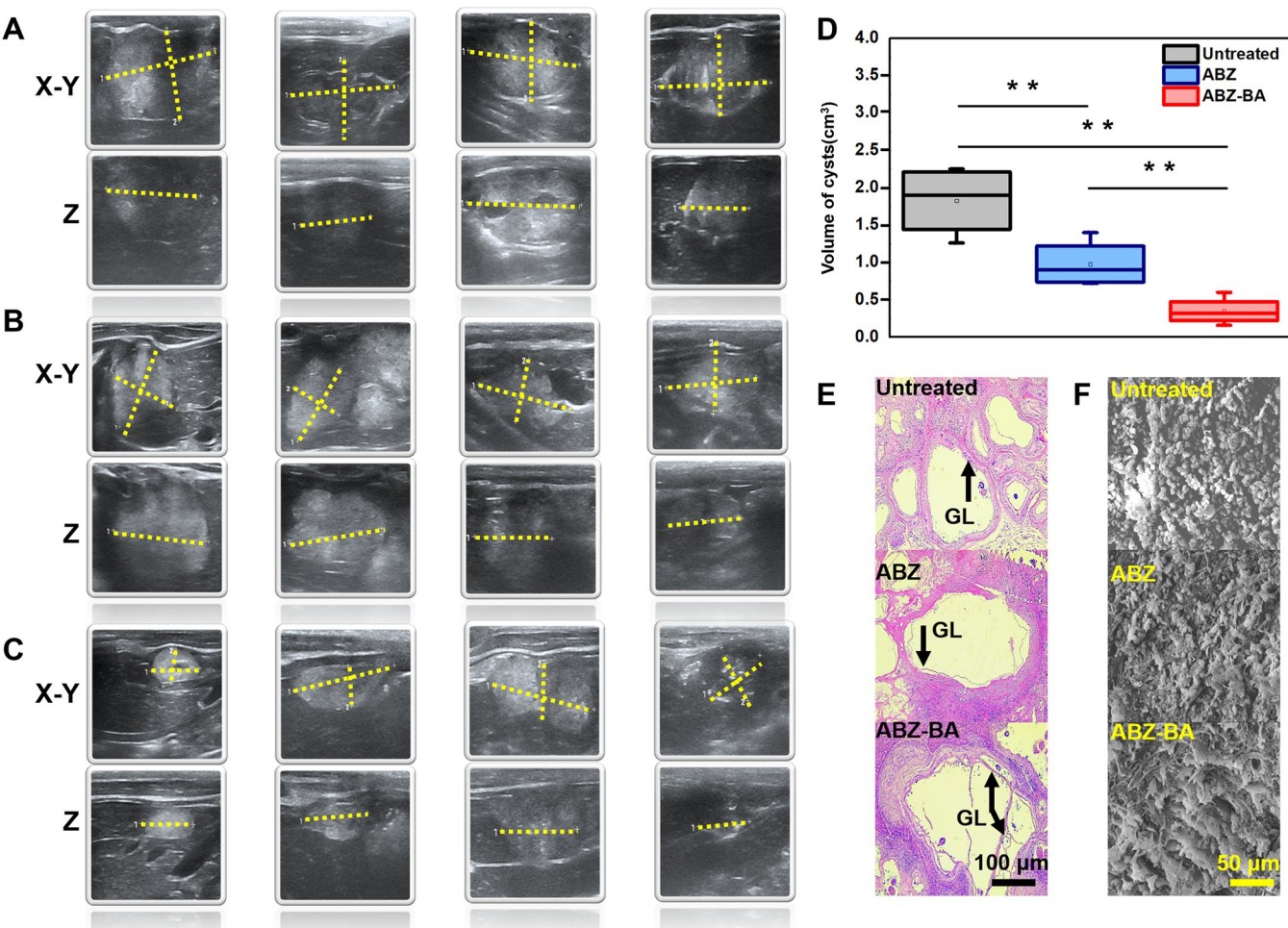

**Fig 7. ABZ-preparation-driven pharmacodynamic analyses in secondary hepatic alveolar rats.** (a)-(c). Cysts were observed in representative liver tissue (n = 6) after 30 days of administration of CMC-Na (a), Albenda (b), and ABZ-BA (c). The X-Y diagram and Z diagram (red) of each group represent the lesion size in the XY-axis and Z-axis directions, respectively. (d). Cysts volume from different treatments (n = 6). Data are expressed as the mean±SD. One-way ANOVA test was used for inter-group comparisons, and Tukey's t-test was used for multiple comparisons. P < 0.05 means the difference is statistically significant(*p<0.05, **p<0.01). (e). Histological images (H&E staining) of cysis. GL, germinal layer. (f). SEM images of cysis after 30d treatment with different formulations.

SEM was used to observe the superficial structure of cyst tissue. In the untreated group, the germinal layer was complete, the infiltrating inflammatory cells were visible, and the cysts were multicellular, with cells that were uniform in size and neatly arranged. In contrast, the encapsulated germinal layer was damaged to varying degrees after ABZ-BA or Albenda treatment (25mg/kg/day, 30 days of administration). The germinal cells in the ABZ-BA treatment group were completely detached, and the structure of the germinal layer was almost completely damaged. The germinal layer in the Albenda treated group remained partially intact and only part of the germinal layer was separated (**Fig 7F**).

There are some limitations to this study. First, the scientific hypothesis put forward includes two scientific hypotheses: the steric hindrance and space-occupying effect of BA could reduce the crystallization of ABZ and improve its solubility. Additionally, specific transport of BA and ASBT could improve the efficiency of ABZ transmembrane transport. We systematically expanded research on reducing the crystallization trend of albendazole from the perspective of

physical pharmaceutics, but for the part of cholic acid promoting transmembrane transport, such as the transmembrane transport mechanism of derivative Caco-2 and intestinal perfusion *in vivo*, many experiments remain in progress and cannot be referenced here. We will produce a systematic report when we have the latest research progress. Second, according to published literature, different linker lengths will affect the physical stability and transmembrane transport efficiency of derivatives, which is another important part of this study. However, due to limited time and manuscript length, the researchers have not elaborated on this in detail and will publish their analysis in another piece of literature.

## Conclusion

As the most effective first-line drug for treating echinococcosis, the poor water solubility of ABZ is the key factor limiting its efficacy. In this study, albendazole bile acid derivatives were prepared with 4-amino-1-butanol as a linker showing superior drug release behavior and anti-*multilocular Echinococcus* effects. Additionally, the introduction of bile acid inhibited the crystallization of drugs by increasing steric hindrance, changing from a crystalline to an amorphous state; on the other hand, transmembrane transport of bile acids by ASBT expressed in the gastrointestinal tract effectively improved the absorption efficiency of ABZ-BA. The synergistic effect of steric hindrance and transmembrane transport increased the equilibrium solubility of ABZ-BA *in vitro* by nearly 4-fold and the relative bioavailability ($F_{rel}$) *in vivo* by about 26-fold. The *in vivo* study demonstrated that the pharmacokinetic behavior of the ABZ-BA was superior to that of Albenda. More importantly, the inhibition test of protoscolex *in vitro* showed that the killing ability of ABZ-BA on protoscolex *in vitro* was the same as that of ABZ. In the pharmacodynamics study, it was also found that the ABZ-BA group had had 2.7-fold greater than that of Albenda after 1 month of oral administration. This albendazole bile acid derivative provides an effective and translatable novel approach to oral drug therapy for chronic, fatal, and neglected parasitic infectious alveolar echinococcosis.

## Supporting information

**S1 Table. Standard curve table of UHPLC-MS method for ABZ-BA and ABZ.**
(XLSX)

**S2 Table. Inter- and intra-day accuracy and precision for the determination of total alkaloid in rat plasma.**
(XLSX)

**S3 Table. Extraction recovery and Matrix effect for the assay of total alkaloid in rat plasma.**
(XLSX)

**S4 Table. The stability for the determination of total alkaloid in rat plasma.**
(XLSX)

**S1 Fig. Graphical table of contents.** An optimized formulation containing Albendazole-bile derivative (ABZ-BA) was developed, which significantly improved the pharmacokinetics and the anti-AE efficacy, after a 30-day, once-daily oral administration.
(TIF)

**S2 Fig. Typical UHPLC-MS chromatogram of plasma.** (A) Blank plasma; (B) Blank plasma +ABZ-BA and ABZ; (C) ABZ-BA and MBZ in plasma after oral administration; (D) ABZ and MBZ in plasma after oral administration.
(TIF)

## Author Contributions

**Conceptualization:** Chunhui Hu.

**Data curation:** Meng Qin, Fabin Zhang, Tao Du.

**Funding acquisition:** Chunhui Hu.

**Methodology:** Meng Qin, Ruixue Gao.

**Writing – original draft:** Xuehui Gan.

**Writing – review & editing:** Chunhui Hu.

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
