## [Decision Letter · Decision Letter 0]

13 Aug 2022

Dear Dr Chunhui Hu ,

Thank you very much for submitting your manuscript "Improve Anti-alveolar Echinococcosis Efficacy by a Novel Albendazole-Bile acids Derivatives with Enhanced Oral Bioavailability" for consideration at PLOS Neglected Tropical Diseases. As with all papers reviewed by the journal, your manuscript was reviewed by members of the editorial board and by several independent reviewers. In light of the reviews (below this email), we would like to invite the resubmission of a significantly-revised version that takes into account the reviewers' comments. 

We cannot make any decision about publication until we have seen the revised manuscript and your response to the reviewers' comments. Your revised manuscript is also likely to be sent to reviewers for further evaluation.

Sincerely,

Alessandra Morassutti, PhD

Academic Editor

Jennifer Keiser

Section Editor

Reviewer's Responses to Questions

**Key Review Criteria Required for Acceptance?**

**Methods**

-Are the objectives of the study clearly articulated with a clear testable hypothesis stated?

-Is the study design appropriate to address the stated objectives?

-Is the population clearly described and appropriate for the hypothesis being tested?

-Is the sample size sufficient to ensure adequate power to address the hypothesis being tested?

-Were correct statistical analysis used to support conclusions?

-Are there concerns about ethical or regulatory requirements being met?

Reviewer #1: - Objectives are clearly stated

- The study design is appropriate

- The population is appropriate

- The sample size is adequate to address the hypothesis

- Correct statistical analysis is used

- There are no ethical concerns

Reviewer #2: (No Response)

**Results**

-Does the analysis presented match the analysis plan?

-Are the results clearly and completely presented?

-Are the figures (Tables, Images) of sufficient quality for clarity?

Reviewer #1: - The analysis relates with the study plan

- Results are clearly stated

- The tables and figures are clear

Reviewer #2: (No Response)

**Conclusions**

-Are the conclusions supported by the data presented?

-Are the limitations of analysis clearly described?

-Do the authors discuss how these data can be helpful to advance our understanding of the topic under study?

-Is public health relevance addressed?

Reviewer #1: - Yes

- No

- This section needs elaboration

- Yes

Reviewer #2: (No Response)

**Editorial and Data Presentation Modifications?**

Reviewer #1: The language has some issues. Appropriate sentence structure is required at many places in the manuscript. The authors should also mention references where required.

Limitations of the study should be stated and there is no comparison of efficacy of this approach with other approaches used for this purpose. Even though, this approach presents a possible route to control the AE, there should be emphasis on its significance and its employability in comparison to the other studies.

Reviewer #2: (No Response)

**Summary and General Comments**

Reviewer #1: The study is of significance as it represents one of the novel approaches for the treatment of AE which is affecting a huge chunk of human population. Data are well presented.

Reviewer #2: Alveolar echinococcosis (AE) is an orphan disease as well as a really neglected disease. Anti-AE therapeutic drugs are generally not a focus for pharmaceutical companies due to the economics or the lack of a viable market, and the drug R&D primarily depends on scientific researchers and NPOs. Although several efforts have been made in the last two decades, the anti-AE drug R&D is proceeding slowly; in contrast, the disease burden is still high. So far, albendazole (ABZ) remains the only drug treatment option, but suffers from the drawback of low oral bioavailability. In this context, discovering new chemical entities for the treatment of Echinococcus multilocularis infections should be encouraged. In the present work, the authors designed an albendazole bile acid derivative (ABZ-BA) with the purpose to improve the bioavailability of the parent drug ABZ through changing the crystal form and employing the sodium dependent bile acid transporter expressed in the small intestine. The study is designed reasonably. The chemical synthesis and structural characterization have been done thoroughly, but the work regarding to anti-parasitic activity as well as PK seems insufficient. The manuscript cannot be accepted unless the following major concerns have been addressed.

Major

1. The major drawback of the study is the lack of a therapeutic efficacy study in experimentally infected mice. According to the “Protoscolex collection and cultivation” section, the authors obviously have the access to E. multilocularis infection animal model. In vivo efficacy study must be accomplished before acceptance. It should be verified that whether the increased bioavailability of ABZ-BA claimed by the authors could translate to improved anti-AE treatment outcome. 

2. The in vitro anti-parasitic activity of ABZ-BA was only evaluated against protoscoleces but not metacestodes, while the latter stage is the primary parasite lesions in the intermediate hosts (e.g., humans and mice). The anti-parasitic ability of ABZ-BA against metacestode vesicles should be determined and compared with the parent drug ABZ to make sure that the structural derivatization did not exert negative impact on the activity.

3. The authors claimed that “sodium dependent bile acid transporter (ASBT) contributes for the enhanced bioavailability of ABZ-BA through effective transport of bile acids”, but no experimental evidence was provided.

Minor

1. The authors should add a statement why chose 4-amino-1-butanol as a linker? Is there a small-scale screening to identify an appropriate linker? If yes, please provide the screening design and results.

2. How many replicates were carried out to evaluate the protoscolecidal activity of ABZ-BA? In the terms of in vitro bioassay, at least three replicates are required.

3. For PK study, the authors should take into consideration the gender difference, please explain why only male rats were employed in the PK study. Also specify the formulations of ABZ and ABZ-BA for oral administration; I suppose both drugs are not water soluble.

4. Have the authors evaluated the toxicity of the prepared ABZ-BA? Sometimes derivatization could result in elevated risk of toxicity. If not, I suggest cytotoxicity against human liver cells should be examined as least, as AE primarily affects the liver.

5. The pharmacokinetic parameters, T1/2, clearance rate, F%, should be added into Table 5. 

6. For Figures 3 and 4, make a clear statement that which kind of statistical analysis was used and add it into the figure legends.

7. For Figures 6, the authors should not include the data of ABZ-BA, because only the plasma concentration of the parent drug ABZ matters for the comparison. The current figure design confuses the readers. Same problem in Figure 3A.

8. Figure 7 is needless, should be incorporated into Figure 1.

9. The authors should provide the product information (cat#, manufacturer, place of production) for all reagents and materials. Some of the information is missing in the Methods section, please check it carefully. 

10. The authors should provide the detailed method of isolation of E. multilocularis protoscoleces, since the protocol is complex, and cannot be simplified as the description “Protoscolex (PSC) of E. multilocularis was (not were) collected aseptically from alveolar echinococcosis obtained from the abdominal cavity of gerbils”. 

11. The authors could put “Method validation of Ultra High-Performance Liquid Chromatography (UHPLC)-MS” section and its related methods into a supporting information file, which obviously is not the focus of the work.

12. A detailed ethic statement is necessary for the study, and makes it an independent section.

13. The written English of the manuscript must be subjected to a professional language service and requires the editor’s extra attention.

PLOS authors have the option to publish the peer review history of their article (what does this mean?). If published, this will include your full peer review and any attached files.

Reviewer #1: No

Reviewer #2: No
---

## [Decision Letter · Decision Letter 1]

15 Nov 2022

Dear Ph.D Hu,

Thank you very much for submitting your manuscript "Improvement of Antialveolar Echinococcosis Efficacy of  Novel Albendazole-Bile acids Derivatives with Enhanced Oral Bioavailability" for consideration at PLOS Neglected Tropical Diseases. As with all papers reviewed by the journal, your manuscript was reviewed by members of the editorial board and by several independent reviewers. The reviewers appreciated the attention to an important topic. Based on the reviews, we are likely to accept this manuscript for publication, providing that you modify the manuscript according to the review recommendations. 

Sincerely,

Alessandra Morassutti, PhD

Academic Editor

Jennifer Keiser

Section Editor

Reviewer's Responses to Questions

**Key Review Criteria Required for Acceptance?**

**Methods**

-Are the objectives of the study clearly articulated with a clear testable hypothesis stated?

-Is the study design appropriate to address the stated objectives?

-Is the population clearly described and appropriate for the hypothesis being tested?

-Is the sample size sufficient to ensure adequate power to address the hypothesis being tested?

-Were correct statistical analysis used to support conclusions?

-Are there concerns about ethical or regulatory requirements being met?

Reviewer #1: (No Response)

Reviewer #2: (No Response)

**Results**

-Does the analysis presented match the analysis plan?

-Are the results clearly and completely presented?

-Are the figures (Tables, Images) of sufficient quality for clarity?

Reviewer #1: (No Response)

Reviewer #2: (No Response)

**Conclusions**

-Are the conclusions supported by the data presented?

-Are the limitations of analysis clearly described?

-Do the authors discuss how these data can be helpful to advance our understanding of the topic under study?

-Is public health relevance addressed?

Reviewer #1: (No Response)

Reviewer #2: (No Response)

**Editorial and Data Presentation Modifications?**

Reviewer #1: (No Response)

Reviewer #2: (No Response)

**Summary and General Comments**

Reviewer #1: The manuscript has greatly improved in terms of clarity, understanding and English language. 

Just a few suggestions are given for correction of the sentences:

L70: Omit ‘of the’

L 75: Echinococcus multilocularis should be in italics

L 85: multilocularis is the name and it should be written in its full form

L 86: humans and animals

L 94-96: reference/s missing

L 97-98: This is repetition of what you have stated in opening paragraph

L 120: Rephrase this sentence for more clarity

L 166: bile instead of bill

L 327: ABZ-BA

Reviewer #2: I am glad to see the authors have made efforts to improve the scientific quality of the manuscript. Before acceptance, there are still a few concerns to be addressed.

1. One of the issues is the newly added “Pharmacodynamic evaluation” section. I suppose all the drug formulations were orally given, as it is the proper way to prove the hypothesis of bioavailability improvement, but the authors gave a statement of administration with “injection”. Please give an explanation.

--How long to initiate the drug administration after infection?

--There is no clear statement of animal housing conditions, animal grouping, and administration period in the method section.

--The description of “pharmacodynamic evaluation” is not a common use, “in vivo therapeutic efficacy in an in-situ AE animal model” would be better.

2. The other concern is the statistical analyses. Comparisons between two groups should use the t-test. For comparisons among multiple groups (≥ 3), one-way ANOVA followed by a reasonable post-hoc test (e.g. Tukey’s) or a nonparametric test (e.g. Kruskal–Wallis) should be used. For example, t-test for the comparison of two groups ABZ and Abz in Fig. 4A, and one-way ANOVA or Kruskal–Wallis test for the three groups in Fig. 7D. The authors should recheck the statistical analysis throughout the manuscript.

3. There is no oral bioavailability F% in Table 1, which is a crucial parameter.

4. The improvement of F% and the reduction of cyst volume should be highlighted in the abstract and author summary.

5. For Fig. 5A, 5D and 5E, replace the legends of Group 1-7 with drug names and concentrations.

6. Change all the mass concentrations to molar concentrations in Figures 4-6 and Table 1, easy to make a comparison of ABZ and ABZ-BA.

7. The language editing requires extra attention of the editors.

PLOS authors have the option to publish the peer review history of their article (what does this mean?). If published, this will include your full peer review and any attached files.

Reviewer #1: No

Reviewer #2: No

Figure Files:

Data Requirements:

Reproducibility:

References

---

## [Decision Letter · Decision Letter 2]

17 Dec 2022

Dear Dr Chunhui Hu, We are pleased to inform you that your manuscript 'Improvement of Antialveolar Echinococcosis Efficacy of  Novel Albendazole-Bile acids Derivatives with Enhanced Oral Bioavailability' has been provisionally accepted for publication in PLOS Neglected Tropical Diseases.

Best regards,

Alessandra Morassutti, PhD

Academic Editor

Jennifer Keiser

Section Editor

Reviewer's Responses to Questions

**Key Review Criteria Required for Acceptance?**

**Methods**

-Are the objectives of the study clearly articulated with a clear testable hypothesis stated?

-Is the study design appropriate to address the stated objectives?

-Is the population clearly described and appropriate for the hypothesis being tested?

-Is the sample size sufficient to ensure adequate power to address the hypothesis being tested?

-Were correct statistical analysis used to support conclusions?

-Are there concerns about ethical or regulatory requirements being met?

Reviewer #1: (No Response)

Reviewer #2: (No Response)

**Results**

-Does the analysis presented match the analysis plan?

-Are the results clearly and completely presented?

-Are the figures (Tables, Images) of sufficient quality for clarity?

Reviewer #1: (No Response)

Reviewer #2: (No Response)

**Conclusions**

-Are the conclusions supported by the data presented?

-Are the limitations of analysis clearly described?

-Do the authors discuss how these data can be helpful to advance our understanding of the topic under study?

-Is public health relevance addressed?

Reviewer #1: (No Response)

Reviewer #2: (No Response)

**Editorial and Data Presentation Modifications?**

Reviewer #1: (No Response)

Reviewer #2: (No Response)

**Summary and General Comments**

Reviewer #1: The article is in much better shape and form now but I would again ask to check the English language and make revisions where required.

Reviewer #2: (No Response)

PLOS authors have the option to publish the peer review history of their article (what does this mean?). If published, this will include your full peer review and any attached files.

Reviewer #1: No

Reviewer #2: No

---

## [Editor Report · Acceptance letter]

28 Dec 2022

Dear Ph.D Hu,

We are delighted to inform you that your manuscript, "Improvement of Antialveolar Echinococcosis Efficacy of  Novel Albendazole-Bile acids Derivatives with Enhanced Oral Bioavailability," has been formally accepted for publication in PLOS Neglected Tropical Diseases.

Best regards,

Shaden Kamhawi

co-Editor-in-Chief

Paul Brindley

co-Editor-in-Chief
